# The influence of psychological capital on employment expectations of vocational undergraduate students: The chain mediating role of active coping style and educational flow experience

**Zerui Huang**[1,2]*, **Ismi Arif Ismail**[3], **Akmar Hayati Ahmad Ghazali**[4], **Jeffrey Lawrence D'Silva**[1], **Haslinda Abdullah**[1], **Zeqing Zhang**[1,2]

**1** Institute for Social Science Studies, Universiti Putra Malaysia, 43400 UPM, Serdang, Selangor, Malaysia, **2** School of Culture and Communication, Guangdong Business and Technology University, Zhaoqing, People's Republic of China, **3** Faculty of Educational Studies, Universiti Putra Malaysia, 43400 UPM, Serdang, Selangor, Malaysia, **4** Faculty of Modern Languages and Communication, Universiti Putra Malaysia, 43400 UPM, Serdang, Selangor, Malaysia

* huangzrio@foxmail.com

## Abstract

As vocational undergraduate students in China face increasing challenges in the job market, understanding the factors that shape their employment expectations is crucial. Psychological capital is considered a key factor influencing students' career outlooks. This study aims to examine the relationship between psychological capital (PC) and employment expectations (EE), while exploring the mediating roles of educational flow experience (EFE) and active coping style (ACS) in this relationship. Based on positive psychology and career development theories, a theoretical model was constructed to understand how psychological capital affects employment expectations through the chain mediation of educational flow experience and active coping style. A sample of 693 vocational undergraduate students (316 males and 377 females) from a university in Guangdong Province participated in the study. Participants completed the Psychological Capital Scale (PCS), the Career Expectation Scale (CES), the Educational Flow Experience Scale (EduFlow-2), and the Coping Style Scale (CSS). Results indicate that higher levels of psychological capital significantly predict stronger employment expectations. Moreover, educational flow experience and active coping style both serve as significant mediators in the relationship between psychological capital and employment expectations, with a chain mediation effect also observed. These findings provide valuable insights into the psychological processes that influence career expectations among vocational undergraduates, highlighting the importance of fostering psychological capital and creating supportive learning environments to improve students' employability. The results offer practical implications for educators and policymakers, suggesting that vocational institutions should integrate strategies to enhance psychological capital, promote educational flow experiences, and support active coping styles to better prepare students for the labor market.

**Data availability statement:** The data under-lying the results presented in this study are available from the Inter-university Consortium for Political and Social Research (ICPSR) database. The dataset can be accessed at the following URL: https://doi.org/10.3886/E216621V1. The project citation is: Huang, Zerui. The influence of psychological capital on employment expectations of vocational undergraduate students: The chain mediating role of active coping style and educational flow experience. Ann Arbor, MI: Inter-university Consortium for Political and Social Research [distributor], 2025-01-26.

**Funding:** The author(s) received no specific funding for this work.

**Competing interests:** The authors have declared that no competing interests exist.

## Introduction

In April 2021, Chinese President Xi Jinping emphasized the need to steadily develop vocational undergraduate education during the National Vocational Education Conference [1]. The Ministry of Education in China outlined a plan to establish a number of applied technology and skill-based higher education institutions, aiming for vocational undergraduate education enrollment to account for no less than 10% of higher vocational education enrollment by 2035 [2]. As an integral part of China's higher education system, vocational undergraduate education focuses on cultivating professionals with advanced applied skills, placing greater emphasis on practical abilities compared to traditional undergraduate education [3]. According to the International Standard Classification of Education (ISCED), vocational undergraduate education corresponds to Level 6, which denotes bachelor's degree-level technical and vocational education [4]. This type of education differs from the academically oriented traditional undergraduate programs, emphasizing skill training directly aligned with industry needs. Degrees and specializations in vocational undergraduate institutions are often closely linked to industry requirements, covering fields such as engineering technology, nursing, information technology, and logistics management. These programs are designed with a stronger focus on hands-on abilities and practical operations, in contrast to the theoretical and research-oriented approach of traditional undergraduate education [5].

However, in China, academic education tends to be highly valued, while vocational education is often perceived as less attractive and considered a form of "secondary education" [6]. Vocational institutions are seen as occupying the lowest tier of the educational hierarchy [7], resulting in the disadvantaged position of vocational undergraduates in the job market. Amid the current complex employment landscape, psychological capital, as a manifestation of individuals' positive psychological resources, has emerged as a crucial factor influencing the employment expectations of vocational undergraduates. It enables individuals to face the pressures and challenges of the job market more positively, fostering higher employment expectations.

Furthermore, due to factors such as China's industrial restructuring [8], slower economic growth [9], and the economic impact of Covid-19 [10], various industries in China are grappling with significant challenges, affecting employment opportunities for university graduates. Additionally, the expansion of higher education in China has led to an increase in the number and proportion of university graduates [11]. Although vocational education is receiving growing attention from the Chinese government [12], the crowding-out effect of top university graduates has intensified the employment pressures faced by students from higher vocational institutions [13]. Vocational graduates tend to experience lower income levels and less career stability compared to graduates from traditional institutions [14]. This disparity often leads to disappointment in their academic qualifications and personal capabilities, ultimately fostering dissatisfaction with society [15]. Nonetheless, focusing solely on external pressures such as economic conditions and the job market is insufficient. The employment expectations of vocational undergraduates are not only shaped by external environments but are also significantly driven by internal psychological mechanisms. Psychological capital, as a positive psychological resource, can help students better cope with employment pressures and support the formation of their employment expectations.

Employment expectations refer to individuals' anticipations regarding their future work and career development before entering the job market [16]. Research indicates that employment expectations can positively influence individuals' decisions to pursue further education [17], suggesting that those with higher employment expectations are more likely to advance their education and enter the job market with enhanced qualifications. Moreover, employment expectations not only impact individuals' mental health [18] but also affect their overall

well-being [19], underscoring their positive role in individuals' lives. Previous studies on employment expectations have primarily focused on individual capabilities [20], leadership qualities [21], the gap between expectations and reality [22–24], and work experience [25], with limited exploration of the internal psychological mechanisms underlying employment expectations. Furthermore, studies on university students' employment expectations have predominantly targeted traditional undergraduates [26] and international students [27], with little attention paid to vocational undergraduates. Since the expansion of China's higher education system in 1999, the number of university graduates has surged annually [28]. In 2022, the number of university graduates in China exceeded 10 million [28]. Meanwhile, China's economic slowdown [29] and a tightening job market have added immense pressure to the employment landscape for university students, particularly for the emerging group of vocational undergraduates.

The objective of this study is to investigate the impact of psychological capital on the employment expectations of vocational undergraduates and further analyze the chain mediating roles of active coping style and educational flow experience in this relationship. Against the backdrop of China's vigorous development of vocational education, this research aims to provide theoretical insights and empirical evidence to enhance the employment expectations of vocational undergraduates, helping them better navigate the current complex employment landscape and improve their standing in the job market.

## Literature review

Psychological capital is a key concept developed within the field of organizational psychology and has since extended to various disciplines, garnering widespread attention from scholars [30]. Generally, psychological capital refers to an individual's positive psychological state of development, encompassing hope, self-efficacy, resilience, and optimism [31]. This study adopts the general definition of psychological capital, positing that psychological capital in vocational undergraduate education reflects the positive psychological state of vocational undergraduates' individual development. Beyond directly impacting mental health, psychological capital has been shown to significantly influence career decision-making [32], employment skills [33], entrepreneurial intentions [34], and job satisfaction [35]. Research indicates that psychological capital can help college students better cope with uncertainties in the job market [36]. In the entrepreneurial domain, psychological capital indirectly influences entrepreneurial intentions by fostering positive coping strategies and problem-solving abilities [37]. Additionally, psychological capital boosts confidence in career choices, reduces employment anxiety, and enhances the effectiveness of career decisions [38]. Psychological capital is broadly associated with college students' employment activities, positively influencing their employment skills [33]. Moreover, it impacts entrepreneurial intentions through traditional capital [39] and directly affects students' perceived employability [40].

Employment expectations are typically shaped by current economic conditions, the availability of job opportunities, and the intensity of labor market competition [41]. However, as a positive psychological resource, psychological capital also influences the formation of employment expectations by enhancing individuals' confidence in the future and their resilience to stress [42]. Studies have shown that psychological capital helps individuals maintain positive career planning under economic pressure [43]. In the context of economic slowdown and a tightening job market, individuals with higher psychological capital are inclined to pursue career development opportunities through positive employment expectations, thereby improving their chances of employment success [44]. Thus, this study proposes Hypothesis 1: Psychological capital positively predicts the employment expectations of vocational undergraduates (H1).

Flow experience, first proposed by psychologist Csikszentmihalyi, is a psychological state in which individuals become fully immersed in an activity, experiencing deep focus and satisfaction from the task itself [45]. Flow experience is characterized by several key features: clear goals, immediate feedback, a balance between skills and challenges, deep concentration, altered perception of time, and intrinsic enjoyment of the activity [46]. In the educational context, this psychological state is recognized as significantly enhancing learners' motivation, engagement, and learning outcomes [47]. For vocational undergraduate education, flow experience is particularly important [48]. This type of education emphasizes practical skill development, with curricula often incorporating project-based learning, hands-on training, and vocational skill exercises. These features create a conducive environment for students to achieve a state of flow [49]. Research suggests that flow experience in vocational education significantly enhances students' engagement and interest in coursework, positively impacting their career development [50].

Flow experience not only positively affects the learning process but also contributes to the development of professional skills and psychological capital [51]. Studies reveal that students in a state of flow exhibit higher learning initiative and goal focus, which are crucial for cultivating vocational undergraduates' professional competencies [52]. Additionally, flow experience boosts students' self-efficacy, thereby enhancing their confidence in future career goals [53]. As a highly immersive psychological state, flow experience strengthens students' interest in learning activities, fostering greater expectations and confidence in their future careers [54]. For vocational undergraduates, flow experience not only enhances learning motivation but also indirectly improves their employment expectations by promoting skill acquisition and goal attainment. Therefore, this study proposes Hypothesis 2: Educational flow experience mediates the relationship between psychological capital and employment expectations of vocational undergraduates (H2).

Coping style refers to the strategies individuals use to alleviate stress and address challenges [55]. Active coping style denotes a strategy whereby individuals actively seek to resolve problems or reduce their negative impact through effort and positive action [56]. For vocational undergraduates, active coping style represents the proactive strategies they adopt to manage stress related to life, study, and employment. Studies have shown that active coping not only positively affects mental health [57] but also enhances individuals' subjective well-being [58]. Furthermore, psychological capital can promote active coping, thereby reducing employment anxiety [38]. Thus, this study proposes Hypotheses 3 and 4: Active coping style mediates the relationship between psychological capital and employment expectations of vocational undergraduates (H3). Active coping style and educational flow experience jointly serve as a chain mediating mechanism between psychological capital and employment expectations of vocational undergraduates (H4).

Based on the study's background and literature review, a research hypothesis is proposed (Fig 1). Educational flow experience and active coping style are identified as key mediating factors in the relationship between psychological capital and employment expectations, with a chain relationship existing between these two mediators.

## Materials and methods

### Participants

With the informed consent of the participants, this study distributed questionnaires online through the professional survey platform Wenjuanxing from April 10 to April 20, 2024, carefully screening participants to ensure quality. The survey targeted undergraduate students from a vocational university in Guangdong Province, aged between 18 and 22 years old (mean age = 19.970, SD = 1.362). Before starting the survey, participants were required to read and

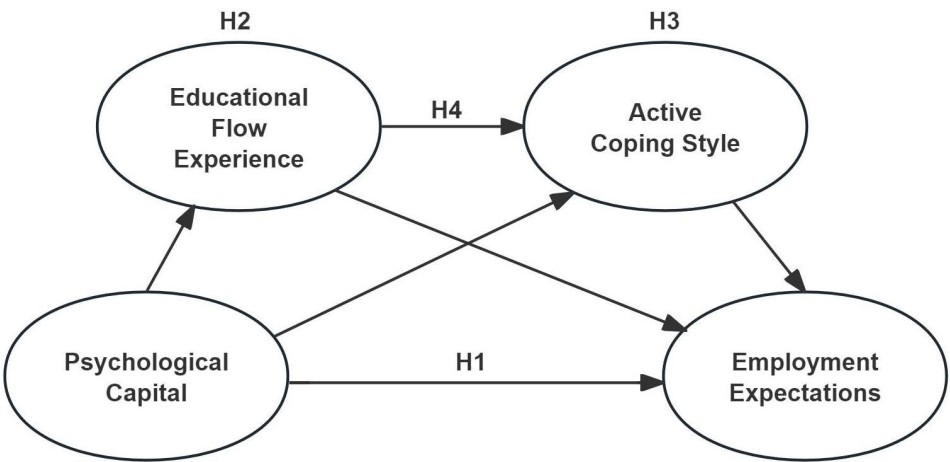

**Fig 1. Hypothesis model.**

agree to an online informed consent form, which detailed the study's purpose, procedures, data confidentiality measures, and potential implications to ensure that participants fully understood and voluntarily participated.

A total of 750 questionnaires were distributed during the survey period. Rigorous screening and quality control measures were applied to eliminate incomplete responses, repetitive answers, and illogical responses. Ultimately, 693 undergraduate students were included in the research sample, resulting in an effective response rate of 92.4%. Among the participants, 54.40% were female (377), and 45.60% were male (316). A respondents' descriptive statistics table is provided in S2 appendix.

To enhance the authenticity and validity of the questionnaire data, several measures were adopted. First, anonymity and data security were ensured through the online platform to reduce participants' concerns and encourage honest responses. Second, strict data screening criteria were followed to ensure the representativeness of the final sample and the high quality of the data.

## Measures

To ensure the applicability of the scales in the Chinese context, all scales used in this study underwent a back-translation process. Initially, the original English scales were translated into Chinese, followed by a reverse translation into English by a separate translator who was not involved in the initial translation. By comparing the two translations and making necessary adjustments, the final Chinese versions of the scales were determined. A summary of scale items, descriptive statistics, and reliability coefficients is provided in S3 appendix.

The confirmatory factor analysis (CFA) was conducted separately for each scale (Psychological Capital, Employment Expectations, Educational Flow Experience, and Active Coping Style) to verify their respective convergent and discriminant validity. This analysis ensured that the items within each scale adequately measured their intended dimensions and that each scale demonstrated satisfactory reliability and validity. A CFA fit indices by scale table is provided in S4 appendix.

## Psychological capital

The Positive Psychological Capital Scale (PCS) [59] was used to measure psychological capital across four dimensions: self-efficacy (7 items), resilience (7 items), optimism (6 items), and

hope (6 items). For example, one item is: Many people admire my talents. This scale has been validated as reliable and valid among college students in mainland China [60]. A 7-point Likert scale was employed, where 7 = strongly agree and 1 = strongly disagree. Items 8, 10, 12, and 14 in the resilience dimension, along with item 25 in the hope dimension, were reverse-scored. Higher total and subscale scores indicated higher levels of psychological capital. The Cronbach's alpha for the four subscales ranged from 0.943 to 0.960. The CFA results were $\chi^2$/df = 2.728, RMSEA = 0.049, CFI = 0.979, NFI = 0.967, and GFI = 0.917.

## Employment expectations

The Career Expectations Scale (CES) [61] was used to measure employment expectations across four dimensions: long-term career success goals (6 items), comfortable work environment and interpersonal relationships (5 items), job attribute preferences (8 items), and career values (6 items). For example, one item is: The job requires originality and creativeness. This scale has been validated as reliable and valid among Chinese college students [62]. A 7-point Likert scale was employed, where 7 = strongly agree and 1 = strongly disagree. Higher total and subscale scores indicated higher levels of employment expectations. The Cronbach's alpha for the four subscales ranged from 0.853 to 0.942. The CFA results were $\chi^2$/df = 2.197, RMSEA = 0.042, CFI = 0.986, NFI = 0.975, and GFI = 0.936.

## Educational flow experience

The Educational Flow Scale (EduFlow-2) [63] was used to measure educational flow experience across four dimensions: cognitive control (3 items), immersion and time transformation (3 items), loss of self-consciousness (3 items), and autotelic experience (3 items). For example, one item is: I trust my ability to meet the high demands of the situation. This scale has been validated as reliable and valid among college students in mainland China [64]. A 7-point Likert scale was employed, where 7 = strongly agree and 1 = strongly disagree. Higher total and subscale scores indicated higher levels of educational flow experience. The Cronbach's alpha for the four subscales ranged from 0.884 to 0.913. The CFA results were $\chi^2$/df = 2.775, RMSEA = 0.048, CFI = 0.989, NFI = 0.983, and GFI = 0.965.

## Active coping style

The Coping Style Scale (CSS) [65] was used to measure active coping style across four dimensions: rational problem-solving (4 items), resigned distancing (4 items), seeking support and ventilation (4 items), and passive wishful thinking (4 items). For example, one item is: I thought about what I would say or do. This scale has been validated as reliable and valid [66]. A 7-point Likert scale was employed, where 7 = strongly agree and 1 = strongly disagree. Items under the resigned distancing and passive wishful thinking dimensions were reverse-scored. Higher total and subscale scores indicated higher levels of active coping style. The Cronbach's alpha for the four subscales ranged from 0.947 to 0.954. The CFA results were $\chi^2$/df = 2.751, RMSEA = 0.049, CFI = 0.986, NFI = 0.979, and GFI = 0.952.

## Statistical analyses

This study employed SPSS 27.0, AMOS 27.0, and their extended plugins for data entry, processing, and statistical analysis. The methods included descriptive statistics, Pearson correlation analysis and PROCESS macro analysis to comprehensively examine the direct, indirect, and chain mediation effects of psychological capital, active coping style, and educational flow experience on employment expectations. To ensure data consistency, all data were standardized using z-scores. Given the self-reported nature of the collected data, potential method

bias was addressed through Harman's single-factor test [67]. To enhance the robustness and accuracy of effect estimation, the study employed the bootstrapping method. Confidence intervals for BootLLCI and BootULCI were set to 95%, and a sample size of 5000 was used for bootstrapping. Mediation effects were deemed significant if the confidence intervals did not include zero [68]. These steps ensured the reliability and accuracy of the findings.

## Ethical considerations

This study received approval from the Ethics Committee of Guangdong Institute of Business and Technology on March 19, 2024 (Approval No. 2024GS032). To protect participants' privacy, the study was conducted anonymously. Prior to participation, detailed informed consent forms were provided to all participants, and written consent was obtained.

## Results

### Common method deviation test

In this study, the psychological capital, employment expectations, educational flow experience, and active coping style were measured using scales, and Harman's single-factor method was applied to test for common method bias. The analysis identified 16 common factors with eigenvalues greater than 1, with the first factor explaining 25.93% of the variance, which is below the 40% threshold, indicating no problematic common method bias [69].

### Descriptive statistics

Table 1 presents the means, standard deviations, and correlation coefficients of the study variables. Psychological capital exhibited moderate correlations with employment expectations (r = 0.597, p < 0.01), educational flow experience (r = 0.380, p < 0.01), and active coping style (r = 0.449, p < 0.01). Employment expectations were moderately correlated with educational flow experience (r = 0.500, p < 0.01) and active coping style (r = 0.511, p < 0.01). A moderate correlation was also observed between active coping style and educational flow experience (r = 0.349, p < 0.01). These results meet the requirements for mediation analysis. Furthermore, all correlation coefficients were below 0.700, indicating the absence of multicollinearity in the data [70].

### Mediating effect

To test the hypotheses, this study employed bootstrapping within a structural equation modeling (SEM) framework to examine the chain mediating model of active coping style and educational flow experience between psychological capital and employment expectations

Table 1. Means, standard deviations and correlation matrices of variables.

|  | 1 | 2 | 3 | 4 |
|---|---|---|---|---|
| 1. Psychological capital | – |  |  |  |
| 2. Employment expectations | .597** | – |  |  |
| 3. Educational flow experience | .380** | .500** | – |  |
| 4. Active coping style | .449** | .511** | .349** | – |
| Mean | 5.223 | 4.892 | 5.732 | 5.345 |
| SD | 0.442 | 0.540 | 0.510 | 0.533 |

Notes: N = 693.

**p < 0.01.

[71]. Table 2 presents the results of the bootstrapping analysis with 95% confidence intervals (5,000 samples), which excluded zero [72]. The direct effect of psychological capital on employment expectations was significant (γ = 0.471, p < 0.001), supporting Hypothesis 1. For the pathway from psychological capital → educational flow experience → employment expectations, the bootstrapping 95% confidence interval (5,000 samples) also excluded zero (γ = 0.124, p < 0.001), supporting Hypothesis 2. For the pathway from psychological capital → active coping style → employment expectations, the bootstrapping 95% confidence interval (5,000 samples) similarly excluded zero (γ = 0.110, p < 0.001), supporting Hypothesis 3. For the pathway from psychological capital → educational flow experience → active coping style → employment expectations, the bootstrapping 95% confidence interval (5,000 samples) again excluded zero (γ = 0.024, p < 0.001), indicating a significant chain mediating effect, and supporting Hypothesis 4.

## Discussion

In this study, we developed a chain mediation model to investigate how psychological capital influences the employment expectations of vocational undergraduate students, focusing on the mediating roles of educational flow experience and active coping style. The findings demonstrate a significant positive relationship between psychological capital and employment expectations, confirming that psychological capital is a strong predictor of higher employment expectations (H1). Further analysis indicates that educational flow experience acts as a mediator in this relationship, suggesting that students with greater psychological capital are more likely to achieve a state of educational flow, which in turn enhances their employment expectations (H2). Similarly, active coping style also mediates the effect of psychological capital on employment expectations, as higher psychological capital promotes more proactive problem-solving approaches, thereby strengthening students' employment aspirations (H3). Moreover, the study reveals that psychological capital influences employment expectations through a sequential mediation process involving both educational flow experience and active coping style (H4), highlighting the interconnected nature of these factors in shaping vocational undergraduates' outlook on their future careers.

### Findings

This study thoroughly explored the impact of psychological capital on the employment expectations of vocational undergraduates, revealing a significant positive predictive effect of psychological capital on employment expectations. This finding supports Hypothesis 1, which posits that the higher the level of psychological capital, the higher the employment

**Table 2. Bootstrapping test result for chain mediating effects.**

| Total, direct, and indirect effect | Effect | Se | Boot95% CI | |
|---|---|---|---|---|
| | | | LLCI | ULCI |
| PC → EFE → EE | 0.124 | 0.029 | 0.069 | 0.182 |
| PC → ACS → EE | 0.110 | 0.019 | 0.075 | 0.146 |
| PC → EFE → ACS → EE | 0.024 | 0.009 | 0.009 | 0.044 |
| Direct effects: PC → EE | 0.471 | 0.039 | 0.395 | 0.546 |
| Total indirect effect | 0.258 | 0.042 | 0.176 | 0.335 |
| Total effect | 0.729 | 0.037 | 0.656 | 0.802 |

Notes: PC, Psychological Capital; EE, Employment expectations; EFE, Educational flow experience; ACS, Active coping style.

expectations of vocational undergraduates. This aligns with existing literature highlighting the positive role of psychological capital in enhancing employment expectations, employability, and job search processes [40,73,74].

Psychological capital comprises four core components: self-efficacy, hope, resilience, and optimism [31]. Individuals with high self-efficacy are more likely to achieve job search success [75], enhancing their positive expectations for future employment. Resilience enables individuals to better adapt to changes and challenges in the job market, such as economic fluctuations and industry adjustments [76]. This adaptability helps vocational undergraduates persist in solving employment challenges rather than giving up easily. Hope and optimism inspire individuals to maintain a positive outlook on their future careers [77], motivating them to exert continuous effort in the job search process. Vocational undergraduates with high psychological capital are more likely to set higher career goals and perform better in job searches, potentially attracting more job opportunities.

The findings further confirmed the independent mediating roles of educational flow experience and active coping style in the relationship between psychological capital and employment expectations, supporting Hypotheses 2 and 3.

First, psychological capital influences employment expectations through educational flow experience. The results indicated that psychological capital significantly and positively predicts educational flow experience, consistent with prior research [78]. Vocational undergraduates with high psychological capital are more confident in mastering new knowledge and skills, making them more likely to experience a state of educational flow. When individuals frequently experience flow in education and learning, they often feel greater pleasure, achievement, and satisfaction [79], enhancing their confidence in their professional abilities. This study confirms that educational flow experience increases employment expectations.

Second, psychological capital influences employment expectations through active coping style. The results showed that psychological capital significantly and positively predicts active coping style, consistent with previous research findings [80]. Vocational undergraduates with high psychological capital believe in their ability to overcome difficulties and challenges, prompting them to adopt proactive problem-solving strategies. The influence of active coping style on employment expectations was also found to be significant and positive, consistent with prior studies [38]. Active coping strategies help vocational undergraduates effectively handle stress and challenges in the job search process, such as interview failures or job rejections, reducing the negative impact of these setbacks on their employment expectations.

The study also revealed that psychological capital influences vocational undergraduates' employment expectations through the chain mediating roles of educational flow experience and active coping style, consistent with previous research [81] and supporting Hypothesis 4. The main effect model of educational flow experience suggests that it not only enhances employment expectations [48] but also positively impacts active coping style [82]. Thus, higher levels of educational flow experience lead to more proactive coping strategies. This demonstrates that educational flow experience positively affects both active coping style and employment expectations, promoting a more positive state of coping.

## Implications

The results of this study indicate that psychological capital has a significant positive impact on the employment expectations of vocational undergraduates, with educational flow experience and active coping style playing key mediating roles. These findings hold important implications for vocational education policies. In the context of China's efforts to promote vocational undergraduate education, these results provide empirical support for developing more effective education policies. Specifically, fostering students' psychological capital can effectively

enhance their employment expectations and competitiveness in the job market, aligning with China's student-centered vocational education philosophy [83].

Vocational institutions should enrich curriculum content, improve teaching methods, and optimize educational resources to provide students with higher-quality learning experiences [84]. For instance, project-based learning and practice-oriented courses [85] can help students achieve high levels of educational flow, enhancing their professional competencies and confidence in employment.

Furthermore, this study offers practical pathways and policy recommendations for vocational institutions. First, it is recommended that national policies establish programs to support the development of students' psychological capital, such as workshops and lectures on enhancing optimism and career resilience [86]. Second, educational institutions should provide more supportive learning environments, including mentorship programs, internships, and industry visits, to help students build self-efficacy and coping skills in real-world settings. Finally, at the instructional level, courses should integrate elements of psychological capital development, such as group collaboration, positive feedback, and case studies, to foster students' hope and resilience. Career planning courses should also incorporate active coping strategies to help students take constructive actions when facing employment challenges.

## Limitations

This study has several limitations: (1) The sample primarily consisted of undergraduate students from a vocational university in Guangdong, which may limit the generalizability of the findings. Future studies could include a broader range of regions and vocational institutions to verify the robustness and applicability of the conclusions. (2) While this study focused on the relationships among psychological capital, educational flow experience, active coping style, and employment expectations, future research could consider other potential influencing factors, such as family capital and personality traits, to provide a more comprehensive understanding of the factors affecting employment expectations.

## Conclusion

This study preliminarily revealed the relationship between psychological capital and employment expectations among vocational undergraduates, as well as the roles of educational flow experience and active coping style. Psychological capital positively predicted employment expectations, while educational flow experience and active coping style played mediating roles in this relationship.

These findings hold significant practical value for improving the employment quality of vocational undergraduates. First, vocational undergraduates can enhance their employment expectations and market adaptability by developing psychological capital, such as improving self-efficacy, fostering optimism, increasing resilience, and maintaining hope. This not only strengthens their intrinsic motivation to face job market challenges but also helps them exhibit greater flexibility and creativity during the job search process.

For university administrators, understanding the role of psychological capital in students' employment preparation is crucial. Administrators should consider integrating psychological capital development into curriculum design and student services, such as offering workshops and lectures to enhance psychological capital and providing career counseling to help students identify and develop constructive coping strategies. Additionally, vocational universities can create supportive learning environments by encouraging students to participate in activities that trigger educational flow, such as project-based learning, internships, and industry visits, which are effective ways to enhance students' educational experiences and psychological capital.

## Supplementary material

**S1 Research Questionnaire.**
(DOCX)

**S2 Appendix. Respondents' Descriptive Statistics.**
(DOCX)

**S3 Appendix. Summary of Scale Items, Descriptive Statistics, and Reliability Coefficients.**
(DOCX)

**S4 Appendix. CFA Fit Indices by Scale.**
(DOCX)

## Acknowledgments

We sincerely appreciate the support and guidance from the teachers who contributed to this research. We are also grateful to the students who took part in the survey, whose participation was essential to the success of this study.

## Author contributions

**Conceptualization:** Jeffrey Lawrence D'Silva.

**Data curation:** Zeqing Zhang.

**Investigation:** Zerui Huang.

**Methodology:** Akmar Hayati Ahmad Ghazali.

**Project administration:** Ismi Arif Ismail.

**Software:** Haslinda Abdullah.

**Supervision:** Ismi Arif Ismail.

**Writing – original draft:** Zerui Huang.

**Writing – review & editing:** Zeqing Zhang.

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
