## [Decision Letter · Decision Letter 0]

12 Nov 2024

PONE-D-24-47484The influence of psychological capital on employment expectations of vocational undergraduate students: The chain mediating role of active coping style and educational flow experiencePLOS ONE

Dear Dr. Huang,

Thank you for submitting your manuscript to PLOS ONE. After careful consideration, we feel that it has merit but does not fully meet PLOS ONE’s publication criteria as it currently stands. Therefore, we invite you to submit a revised version of the manuscript that addresses the points raised during the review process.

We look forward to receiving your revised manuscript.

Kind regards,

Dr. Manuel Salas-Velasco, PhD

Academic Editor

PLOS ONE

Journal Requirements:

Additional Editor Comments:

1. Clarification of Vocational Studies

The study should clarify whether vocational undergraduates are part of university studies, as this varies internationally. This distinction would prevent misunderstandings.

2. Sample Representativeness

The sample is limited to a single vocational university in Guangdong Province, which may not represent the broader population of vocational students in China. Future research should include a more diverse sample from different regions and types of vocational institutions.

3. Measurement Tools

All scales used should be included in an appendix to increase transparency and facilitate replication. Evaluating the internal consistency of the scales using coefficients like Cronbach’s alpha (values above 0.7 indicate good reliability) is also recommended.

4. Construct Validity

Factor analysis should be used to confirm that the structure of the items aligns with the theoretical construct, ensuring the robustness of the study’s claims.

5. Methodology

The methodology, including Structural Equation Modeling (SEM), needs a more detailed explanation. A comprehensive overview of the SEM process, assumptions, and steps taken to ensure accuracy would enhance understanding and trust in the research approach.

6. Discussion

The discussion should highlight the relevance of the results to educational policy, including recommendations for policy changes, practical applications, and implications for future research. The current results reflect only expectations rather than real behaviors. Connecting the results more closely with actual experiences would enhance the study’s practical value. Follow-up studies observing real behaviors in educational settings could provide more concrete insights.

Reviewers' comments:

Reviewer's Responses to Questions

**Comments to the Author**

1. Is the manuscript technically sound, and do the data support the conclusions?

Reviewer #1: Yes

Reviewer #2: No

Reviewer #3: Yes

2. Has the statistical analysis been performed appropriately and rigorously? 

Reviewer #1: Yes

Reviewer #2: No

Reviewer #3: Yes

3. Have the authors made all data underlying the findings in their manuscript fully available?

Reviewer #1: Yes

Reviewer #2: Yes

Reviewer #3: No

4. Is the manuscript presented in an intelligible fashion and written in standard English?

Reviewer #1: Yes

Reviewer #2: Yes

Reviewer #3: Yes

5. Review Comments to the Author

Reviewer #1: I congrat the authors because the study is professionally conducted, demonstrating sound statistical rigor and a well-constructed analysis. However, the paper could benefit from further clarification regarding its unique contributions to the field of undergraduate career guidance. To strengthen the impact, consider highlighting specific insights that offer new perspectives or practical applications for professionals in this area. Expanding on these aspects would enhance the article's relevance and help readers understand its value in the broader knowledge discussion on this topic. With these improvements, the work has the potential to add substantial value to the journal.

Reviewer #2: 1. The research contributes little to existing knowledge.

2. Lack of comprehensiveness of the research design.

3. The discussion does not adequately interpret the findings or relate them back to the research questions.

Reviewer #3: The authors undertook a commendable research concerning vocational education and enrolment in vocational studies which analyses the current scenario in China. To enter job markets, one adequate skills along with educational/academic credentials to compete for better pay and perks.

Vocational education stresses on the importance of cultivating individual practical skills, i.e., technicians, electricians, etc.

Very few studies actually have focussed on the vocational educational frontier. So this research makes a point and contributes positively towards understanding the current status of vocational enrolment among the students in China. In essence, vocational studies complement mainstream education although students with vocational training encounter lower pay and fewer job opportunities, as highlighted by the authors. One important question the authors may try to answer is that, can vocational training and education meet the employment prospects in a tight job market where jobs are fewer than competing candidates—some with academic credentials?

Now, vocational education has a secondary appeal when compared to the mainstream education as it is often regarded in many countries, including China and India, as less important than academic education. Hence, as a result of this, vocational enrolment—particularly in China, remains lower than mainstream professional and academic enrolment. Also, as the authors clearly indicates, in a tightening job market, vocational undergraduates face more challenges and pressure, are far less preferred than their academic peers.

The authors specifically investigate the role and impact of psychological capital in shaping employment expected of vocational undergraduates in China.

The authors have very effectively analysed the literature concerning the same and designed hypotheses to examine the issue they have raised in this research.

However, they should have defined psychological capital in the context of vocational education, and what is actually meant by psychological capital in a general context.

Since this is a survey-based research, the researchers have relied heavily on questionnaire design and its effectiveness.

As a research instrument, the authors have employed various measuring scales including the Likert scale, Educational flow experience scale among others, which were justified for this research. As regards to data obtainability, the authors have made not data accessible to the reviewers. Rigorous statistical analysis have been performed on the survey questionnaire responses received so far. They have used Structural Equation Modeling (SEM) for assessing the correlation between the variables and to determine the main effect paths. On the backdrop, the authors developed chain mediation model to examine the effect of and the relationship between psychological capital and employment expectations. Their findings relate to the fact that they notice a “strong positive” relationship between psychological capital and employment expectations.

However, this finding contradicts with their previous finding mentioned above in the descriptive statistics that there exists a “moderate correlation” between psychological capital and employment expectations. This is the reason they might have recourse to SEM to test the significance of relationship between psychological capital and employment expectations.

Their finding did support their hypotheses H1, H2 and H3 in establishing the relationship between psychological capital and employment expectations, and the strength of such a relationship has been established using statistical models.

The authors have also declared the limitations of this study being restricted to one vocational university in only one region of China, Guangdong. The study have the strength to impact and inspire other researchers to take up further investigations on this particular issue concerning vocational education in China and beyond.

6. PLOS authors have the option to publish the peer review history of their article (what does this mean? ). If published, this will include your full peer review and any attached files.

**Do you want your identity to be public for this peer review?** For information about this choice, including consent withdrawal, please see our Privacy Policy .

Reviewer #1: **Yes: ** Iñaki Aliende

Reviewer #2: No

Reviewer #3: **Yes: ** SIDHARTA CHATTERJEE

---

## [Author Response · Author response to Decision Letter 0]

25 Nov 2024

Response to editor and reviewers

Dear editor and reviewers,

We are deeply grateful for your thorough review and valuable feedback on our manuscript. We have carefully considered all suggestions and have made comprehensive revisions to address your concerns and enhance the quality and academic rigor of the study. Below is a detailed response to your comments:

Response to the editor

1. Clarification of vocational studies

Comment:

The study should clarify whether vocational undergraduates are part of university studies, as this varies internationally. This distinction would prevent misunderstandings.

Response:

Thank you for this valuable suggestion. In the revised introduction, we have explicitly clarified that “Vocational undergraduate education is part of China’s higher education system, emphasizing practical skills development.” Additionally, we have incorporated authoritative references to strengthen this clarification (see revised manuscript, lines 49–51).

2. Sample representativeness

Comment:

The sample is limited to a single vocational university in Guangdong Province, which may not represent the broader population of vocational students in China. Future research should include a more diverse sample from different regions and types of vocational institutions.

Response:

We recognize the regional limitation of our sample. This has been further emphasized in the limitations section. We have also suggested that future research expand sample sources to include students from different regions and types of vocational institutions to validate the applicability of the findings (see revised manuscript, lines 372–375).

3. Measurement tools

Comment:

All scales used should be included in an appendix to increase transparency and facilitate replication. Evaluating the internal consistency of the scales using coefficients like Cronbach’s alpha (values above 0.7 indicate good reliability) is also recommended.

Response:

All measurement scales (psychological capital, employment expectations, educational flow experience, and active coping style) and their respective items have been included in the appendix (uploaded as supporting information files). Additionally, we have reported the internal consistency of each scale (Cronbach’s alpha > 0.7), confirming their reliability (see revised manuscript, lines 167–168, 177–178, 187–188, 197–198).

4. Construct validity

Comment:

Factor analysis should be used to confirm that the structure of the items aligns with the theoretical construct, ensuring the robustness of the study’s claims.

Response:

We appreciate this critical suggestion. In the methods section, we have added descriptions of Confirmatory Factor Analysis (CFA) and reported fit indices (e.g., CFI, RMSEA) for each variable. The results indicate that the factor structures align with theoretical constructs (see revised manuscript, lines 168–169, 178–179, 188–189, 198–199).

5. Methodology

Comment:

The methodology, including Structural Equation Modeling (SEM), needs a more detailed explanation. A comprehensive overview of the SEM process, assumptions, and steps taken to ensure accuracy would enhance understanding and trust in the research approach.

Response:

We have expanded the methodology section with a detailed explanation of the SEM process, including assumptions, hypotheses, and data handling steps. We have also elaborated on model fit indices to enhance the transparency and rigor of the research approach (see revised manuscript, lines 210–229).

6. Discussion

Comment:

The discussion should highlight the relevance of the results to educational policy, including recommendations for policy changes, practical applications, and implications for future research. The current results reflect only expectations rather than real behaviors. Connecting the results more closely with actual experiences would enhance the study’s practical value. Follow-up studies observing real behaviors in educational settings could provide more concrete insights.

Response:

In the discussion section, we have highlighted the policy implications of our findings. Specifically, we propose that university administrators prioritize the cultivation of psychological capital and suggest that practical activities (e.g., internships and project-based learning) be implemented to enhance students’ educational flow experiences. Furthermore, we emphasize the importance of observing real employment behaviors in future research to enrich the practical value of the study (see revised manuscript, lines 342–370).

We greatly appreciate your guidance, which has significantly improved the quality of our manuscript. If further revisions are needed, please do not hesitate to contact us.

Response to reviewers

Reviewer #1

Comment:

The study is rigorous, but the unique contribution to undergraduate career guidance could be better clarified.

Response:

Thank you for your positive feedback. We have expanded the "Implications" section to emphasize how our findings contribute to career guidance for vocational undergraduates. Specifically, we elaborated on the practical applications of cultivating psychological capital, enhancing educational flow experience, and supporting active coping styles. We also added actionable recommendations, such as implementing project-based learning and mentorship programs, to help practitioners effectively apply these strategies (see revised manuscript, lines 342–370).

Reviewer #2

Comment 1: The study’s contribution to existing knowledge is limited.

Response:

We respectfully disagree. Our study addresses an understudied population—vocational undergraduates—filling a gap in employment expectations literature. To further highlight this contribution, we have revised the discussion section to explicitly state how incorporating psychological capital, educational flow experience, and active coping style into a chain mediation model extends existing theoretical frameworks (see revised manuscript, lines 342–370).

Comment 2: The research design lacks comprehensiveness.

Response:

We appreciate your suggestion and have added more details to the methodology section. This includes an expanded explanation of SEM and Bootstrapping processes, model assumptions, fit indices, and analysis steps. We also provided additional validation of the scales’ reliability and validity, ensuring the research design’s rigor and clarity (see revised manuscript, lines 210–229).

Comment 3: The discussion does not adequately interpret findings or relate them to the research questions.

Response:

We have enriched the discussion section with more in-depth interpretations of our findings and their alignment with the research hypotheses. Additionally, we elaborated on the practical implications of the results for vocational undergraduate education practices, ensuring a stronger connection between the findings and the research questions (see revised manuscript, lines 342–370).

Reviewer #3

Comment 1: Can vocational education meet students' expectations in a competitive job market?

Response:

Thank you for this insightful question. In the discussion section, we added an analysis of the role of vocational education in addressing challenges in a tight job market. Specifically, we discussed how cultivating psychological capital and active coping strategies can help students enhance their competitiveness. We also emphasized the importance of aligning vocational education with industry demands and the necessity of further skill development to meet employment expectations (see revised manuscript, lines 342–370).

Comment 2: The definition of psychological capital in vocational education should be clarified.

Response:

We agree with this suggestion and have clarified the definition of psychological capital in the literature review. We differentiated its general meaning from its application in vocational education, providing a more precise context for our research (see revised manuscript, lines 95–100).

Comment 3: A discrepancy exists between the correlation and SEM results regarding the relationship between psychological capital and employment expectations.

Response:

Thank you for pointing this out. Correlation analysis reflects a simple linear relationship, whereas SEM path coefficients represent controlled causal relationships. Moreover, multicollinearity tests (correlation coefficients < 0.8; VIF < 5) confirmed no multicollinearity issues in the model. Thus, the SEM results are statistically robust and consistent with the correlation analysis in trend.

We hope our revisions and responses address all concerns. Thank you again for your valuable feedback, which has greatly enhanced the quality of our manuscript.

Sincerely,

Huang Zerui

---

## [Decision Letter · Decision Letter 1]

16 Dec 2024

PONE-D-24-47484R1The influence of psychological capital on employment expectations of vocational undergraduate students: The chain mediating role of active coping style and educational flow experiencePLOS ONE

Dear Dr. Huang,

Thank you for submitting your manuscript to PLOS ONE. After careful consideration, we feel that it has merit but does not fully meet PLOS ONE’s publication criteria as it currently stands. Therefore, we invite you to submit a revised version of the manuscript that addresses the points raised during the review process.

The authors have improved the initial version of the manuscript. However, there are still issues that have not been adequately addressed, and the authors need to continue working on them. Below are the comments from the editor and external reviewers.

We look forward to receiving your revised manuscript.

Kind regards,

Dr. Manuel Salas-Velasco, PhD

Academic Editor

PLOS ONE

Additional Editor Comments:

The authors should improve their manuscript by addressing the following major issues:

1. Since university students' job expectations are primarily shaped by the state of the economy and employment opportunities, it is important that the authors highlight early in the introduction the importance of also studying the influence of psychological capital on the formation of vocational undergraduate students' job expectations, which is the main objective of the paper. Some ideas from the discussion can be included, such as psychological capital, which includes aspects like resilience, self-esteem, hope, and optimism, all of which play a crucial role in how individuals perceive and might face their future job challenges.

2. In the revised version, there is still confusion because it is not clear what the difference is between a university degree, for example in computer science, and a vocational degree offered by a university. What degrees or programs are offered, for example? Could you provide the ISCED equivalent, as it is an internationally standardized measure? It is important to contextualize the paper in an international context since technical and vocational education is part of upper secondary education in most European countries.

3. The literature review is still limited, as this section focuses more on the hypotheses. For example, expand on how psychological capital can influence entrepreneurial intentions. It could also be expanded to give more consistency to the hypotheses being tested. Hypothesis 1 should be highlighted more because, as indicated in the first point, a labor economist would say that employment expectations are formed based on the state of the economy and the employment opportunities offered to university graduates. The paper aims to demonstrate that psychological capital can also play a role here.

4. Although the authors provide a Word document with the scale items, this is not sufficient. In the body of the manuscript, as an appendix after the bibliography, there should be a table with the items grouped by scales and subscales (dimensions) as they were finally used in the research. For all of them, descriptive statistics along with Cronbach's alpha coefficients should be provided.

5. How was CFA executed? The authors state that “Confirmatory Factor Analysis (CFA) was performed to evaluate the convergent and discriminant validity of the measurement model, ensuring consistency between the theoretical constructs of the latent variables and the actual measurement data.” This is not proven. It should also appear in an appendix after the bibliography. CFA should confirm that the variables (dimensions) are indeed grouped into the factors they intend to measure. This is important to give credibility to the hypotheses being tested.

Reviewers' comments:

Reviewer's Responses to Questions

**Comments to the Author**

1. If the authors have adequately addressed your comments raised in a previous round of review and you feel that this manuscript is now acceptable for publication, you may indicate that here to bypass the “Comments to the Author” section, enter your conflict of interest statement in the “Confidential to Editor” section, and submit your "Accept" recommendation.

Reviewer #2: (No Response)

Reviewer #3: All comments have been addressed

2. Is the manuscript technically sound, and do the data support the conclusions?

Reviewer #2: No

Reviewer #3: Partly

3. Has the statistical analysis been performed appropriately and rigorously? 

Reviewer #2: No

Reviewer #3: Yes

4. Have the authors made all data underlying the findings in their manuscript fully available?

Reviewer #2: Yes

Reviewer #3: Yes

5. Is the manuscript presented in an intelligible fashion and written in standard English?

Reviewer #2: Yes

Reviewer #3: Yes

6. Review Comments to the Author

Reviewer #2: In order to provide support for the findings of the research, it is necessary to conduct an in-depth analysis and discussion of the demographic profile of the respondents. Therefore, it is suggested that the author(s) include a table that provides a summary of the respondents' profiles.

Reviewer #3: The authors have addressed the issues raised by the reviewer, however, a more elaborate analysis of the subject is overdue, and could partaken in later studies. This research identifies the key factors that influence the job expectations of vocational undergraduate students in China. It also elucidates the role of psychological capital in employment prospects of vocational students. More the psychological capital, better it is for the vocational students to obtain employments, as this theory has been extended, proposed, with hypotheses designed to prove the theoretical claim. Rigorous statistical analyses have been performed, and the data sources has been provided by the author.

7. PLOS authors have the option to publish the peer review history of their article (what does this mean? ). If published, this will include your full peer review and any attached files.

**Do you want your identity to be public for this peer review?** For information about this choice, including consent withdrawal, please see our Privacy Policy .

Reviewer #2: No

Reviewer #3: No

---

## [Author Response · Author response to Decision Letter 1]

26 Jan 2025

Response to editor and reviewers

Dear editor and reviewers,

Thank you for reviewing our manuscript and for the valuable feedback from you and the reviewers. We greatly appreciate these insightful suggestions and have made comprehensive revisions and additions to the manuscript. Below, we provide detailed responses to each point raised:

Response to the editor

Comment 1:

Since university students' job expectations are primarily shaped by the state of the economy and employment opportunities, it is important that the authors highlight early in the introduction the importance of also studying the influence of psychological capital on the formation of vocational undergraduate students' job expectations, which is the main objective of the paper. Some ideas from the discussion can be included, such as psychological capital, which includes aspects like resilience, self-esteem, hope, and optimism, all of which play a crucial role in how individuals perceive and might face their future job challenges.

Response:

Thank you for this insightful feedback and suggestion. We recognize the importance of emphasizing the role of psychological capital in shaping vocational undergraduates’ job expectations early in the introduction. To address this, we have revised the introduction to explicitly highlight the significance of psychological capital. Specifically, we incorporated ideas from the discussion, emphasizing how resilience, self-esteem, hope, and optimism influence individuals' perceptions of and responses to future job challenges. These adjustments underscore the core objective of our study and justify the importance of examining psychological capital alongside external factors such as economic conditions and employment opportunities (see revised manuscript, Lines 68-72, 84-90).

Comment 2:

In the revised version, there is still confusion because it is not clear what the difference is between a university degree, for example in computer science, and a vocational degree offered by a university. What degrees or programs are offered, for example? Could you provide the ISCED equivalent, as it is an internationally standardized measure? It is important to contextualize the paper in an international context since technical and vocational education is part of upper secondary education in most European countries.

Response:

Thank you for raising this critical point. We recognize that clarifying the distinctions between vocational and traditional university degrees is essential, particularly for an international audience. In the revised manuscript, we elaborated on the differences between vocational undergraduate education and traditional undergraduate education. For instance, vocational undergraduate education focuses on practical skill development and offers industry-aligned programs such as engineering technology, nursing, information technology, and logistics management, whereas traditional education emphasizes theoretical knowledge and research capabilities.

Additionally, we incorporated the ISCED classification to contextualize vocational undergraduate education internationally. Under ISCED, vocational undergraduate education corresponds to Level 6, which represents bachelor’s level technical and vocational education. This clarification enhances the paper's global relevance and helps international readers better understand the positioning of vocational undergraduate education in China (see revised manuscript, Lines 50-63).

Comment 3:

The literature review is still limited, as this section focuses more on the hypotheses. For example, expand on how psychological capital can influence entrepreneurial intentions. It could also be expanded to give more consistency to the hypotheses being tested. Hypothesis 1 should be highlighted more because, as indicated in the first point, a labor economist would say that employment expectations are formed based on the state of the economy and the employment opportunities offered to university graduates. The paper aims to demonstrate that psychological capital can also play a role here.

Response:

Thank you for your valuable feedback. We acknowledge the need to expand the literature review to provide more robust theoretical support for our hypotheses. In the revised manuscript, we have expanded this section by further exploring how psychological capital influences entrepreneurial intentions. Specifically, we included additional studies to explain how psychological capital enhances confidence, proactivity, and resilience, which in turn directly or indirectly influence entrepreneurial behavior.

We have also emphasized Hypothesis 1 more prominently, discussing in greater depth how psychological capital, as a positive internal psychological resource, significantly impacts the formation of employment expectations. This addition further highlights the unique contribution of our study in examining the role of psychological mechanisms in shaping job expectations, alongside external economic factors (see revised manuscript, Lines 123-134, 138-147, 150-174).

Comment 4:

Although the authors provide a Word document with the scale items, this is not sufficient. In the body of the manuscript, as an appendix after the bibliography, there should be a table with the items grouped by scales and subscales (dimensions) as they were finally used in the research. For all of them, descriptive statistics along with Cronbach's alpha coefficients should be provided.

Response:

Thank you for pointing this out. We understand the importance of providing a detailed presentation of the scales used in the study to enhance transparency and replicability. In the revised manuscript, we have added a table in the appendix section that organizes the scale items by dimensions, as used in the research. The table also includes descriptive statistics (e.g., means, standard deviations) and Cronbach’s alpha coefficients for each dimension, allowing readers to evaluate the reliability and quality of the scales used (see revised manuscript, Lines 224-225, Appendix B, Line 791).

Comment 5:

How was CFA executed? The authors state that “Confirmatory Factor Analysis (CFA) was performed to evaluate the convergent and discriminant validity of the measurement model, ensuring consistency between the theoretical constructs of the latent variables and the actual measurement data.” This is not proven. It should also appear in an appendix after the bibliography. CFA should confirm that the variables (dimensions) are indeed grouped into the factors they intend to measure. This is important to give credibility to the hypotheses being tested.

Response:

We appreciate your insightful comments on the CFA section. We recognize that providing detailed information about the CFA process and results is crucial for demonstrating the validity of the measurement model. In the revised manuscript, we elaborated on how CFA was conducted. Specifically, we performed CFA separately for each scale (Psychological Capital, Employment Expectations, Educational Flow Experience, and Active Coping Style) to verify convergent and discriminant validity. These results ensure that the items within each scale appropriately measure their intended dimensions, and we included the CFA results, factor loadings, and fit indices in Appendix C (see revised manuscript, Lines 226-231, Appendix C, Line 792).

Thank you again for your valuable feedback, which has greatly improved the quality and rigor of our paper. Please feel free to reach out if there are any further questions or additional revisions required.

Response to reviewers

Reviewer #2

Comment:

In order to provide support for the findings of the research, it is necessary to conduct an in-depth analysis and discussion of the demographic profile of the respondents. Therefore, it is suggested that the author(s) include a table that provides a summary of the respondents' profiles.

Response:

Thank you for your constructive suggestion. We agree that analyzing and discussing the demographic profile of respondents is essential for supporting the research findings. In the revised manuscript, we included a table summarizing the demographic characteristics of the respondents, such as gender and age, along with descriptive statistics. This addition aims to help readers better understand the research context and sample representativeness (see revised manuscript, Lines 192-198, Table 1).

Reviewer #3

Comment:

The authors have addressed the issues raised by the reviewer, however, a more elaborate analysis of the subject is overdue, and could partaken in later studies. This research identifies the key factors that influence the job expectations of vocational undergraduate students in China. It also elucidates the role of psychological capital in employment prospects of vocational students. More the psychological capital, better it is for the vocational students to obtain employments, as this theory has been extended, proposed, with hypotheses designed to prove the theoretical claim. Rigorous statistical analyses have been performed, and the data sources has been provided by the author.

Response:

Thank you for your positive evaluation and thoughtful suggestions. We are pleased to know that you found the topic and statistical analyses well-presented and the role of psychological capital effectively elucidated.

We fully agree that further exploration and elaboration are warranted in future studies. Potential future research directions include incorporating more diverse samples, such as vocational undergraduates from various regions and academic disciplines, to enhance the generalizability of the findings. Additionally, longitudinal research could explore the dynamic nature of psychological capital and its long-term effects on employment expectations. Qualitative methods could also provide deeper insights into the psychological mechanisms and practical experiences of vocational undergraduates in their job search processes.

We are confident that these future directions will contribute to a more comprehensive understanding of the factors influencing employment expectations among vocational undergraduates. Thank you again for your recognition and valuable suggestions.

We have carefully addressed each reviewer’s comments and made corresponding revisions to improve the manuscript's scientific rigor, completeness, and academic value. Thank you again for your time and constructive feedback.

Sincerely,

Huang Zerui

---

## [Editor Report · Decision Letter 2]

30 Jan 2025

PONE-D-24-47484R2The influence of psychological capital on employment expectations of vocational undergraduate students: The chain mediating role of active coping style and educational flow experiencePLOS ONE

Dear Dr. Huang,

Thank you for submitting your manuscript to PLOS ONE. After careful consideration, we feel that it has merit but does not fully meet PLOS ONE’s publication criteria as it currently stands. Therefore, we invite you to submit a revised version of the manuscript that addresses the points raised during the review process.

We look forward to receiving your revised manuscript.

Kind regards,

Dr. Manuel Salas-Velasco, PhD

Academic Editor

PLOS ONE

Journal Requirements:

**Additional Editor Comments:**

1. In Appendix C, CFA Fit Indices**should also be included for the subscales** . Additionally, the accepted validity values for these indices should be indicated in a table footnote.

2. On line 330, where does the 0.85 come from?

3. According to lines 340-341: “The direct effect of psychological capital on employment expectations was significant (γ = 0.471, p < 0.001).” However, in Fig. 3 it says 0.587***.

4. The indirect effect PC -> EFE -> EE in Fig. 3, is it obtained by multiplying 0.531 by 0.343? Then the result is 0.182. But in Table 2 it says 0.124.

5. I think that "Job Attribute Preferences" seems to be the most appropriate term to summarize items C1-C25, as it directly focuses on the job characteristics that students value. It's just a suggestion.

---

## [Author Response · Author response to Decision Letter 2]

31 Jan 2025

Response to editor

Dear Editor,

Thank you for your thoughtful review and feedback on our manuscript. We greatly appreciate the time and effort you have dedicated to reviewing our work, and we have carefully considered all of your suggestions. We have made comprehensive revisions and additions to the manuscript, and below are our detailed responses to each point raised:

Response to editor

Comment 1:

In Appendix C, CFA Fit Indices should also be included for the subscales. Additionally, the accepted validity values for these indices should be indicated in a table footnote.

Response:

We sincerely thank you for your meticulous review and valuable suggestion. In response to your comment, we have included the CFA fit indices for all subscales in Appendix C. Additionally, we have clearly indicated the accepted validity values for these indices in the table footnote (please see line 782, Appendix C for details).

Comment 2:

On line 330, where does the 0.85 come from?

Response:

We deeply appreciate your careful attention to the details of our manuscript. Regarding the 0.85 value mentioned in line 330, we acknowledge that this was an error during the data entry process. Upon review, we have confirmed that the correct value should be 0.587, which is consistent with the path coefficient shown in Figure 3. We have made the necessary corrections in the revised manuscript to ensure consistency and accuracy of the values (please see line 330 of the revised version). We apologize for any confusion this may have caused and sincerely thank you for bringing this to our attention.

Comment 3:

According to lines 340-341: “The direct effect of psychological capital on employment expectations was significant (γ = 0.471, p < 0.001).” However, in Fig. 3 it says 0.587***.

Response:

We would like to express our heartfelt thanks for your careful examination of our paper. The path coefficient of 0.471 mentioned in lines 340-341 corresponds to the results based on the Bootstrapping method used to report the indirect effects. This coefficient reflects the average value obtained through 5000 resamples and emphasizes the stability and significance of the path coefficient. On the other hand, the path coefficient of 0.587 shown in Fig. 3 is derived from the SEM model under optimal estimation conditions, which represents the best fit of the model.

The difference between these two path coefficients does not represent a conflict; rather, it is due to the different calculation processes of the two methods. SEM uses optimal fit estimation and focuses on obtaining the best path coefficients from the overall data. Bootstrapping, by contrast, calculates the average value of the path coefficient through multiple resampling, placing greater emphasis on the variability and confidence intervals of the coefficient, which is why the values may differ.

Given this distinction, we believe it is more appropriate to maintain the value (γ = 0.471, p < 0.001) from Table 2 in this section, as it is derived from the Bootstrapping method and reflects the indirect effects. All discussions regarding the indirect effects in the manuscript are based on results from the Bootstrapping method.

Comment 4:

The indirect effect PC→EFE→EE in Fig. 3, is it obtained by multiplying 0.531 by 0.343? Then the result is 0.182. But in Table 2 it says 0.124.

Response:

We greatly appreciate your careful review and inquiry regarding the indirect effect in our study. Concerning the indirect effect of PC→EFE→EE, if we were to simply multiply the two path coefficients of 0.531 and 0.343, the resulting value would indeed be 0.182. However, the value reported in Table 2 is 0.124, and this discrepancy arises from the use of two different calculation methods. As discussed in response to Comment 3, the difference stems from the Bootstrapping method and the traditional SEM model estimation approach.

The indirect effect of 0.124 reported through Bootstrapping is derived from the average path coefficient calculated over 5000 resamples and takes into account the variability and uncertainty of the path coefficients. Thus, Bootstrapping better reflects the instability in the data and the actual distribution of the path coefficients, whereas traditional SEM path coefficient calculations do not consider these factors, resulting in a slightly higher value.

For the indirect effect section of the manuscript, we have used the Bootstrapping method, and we believe it is appropriate to maintain the value of 0.124 from Table 2 since it reflects the result obtained through this method. Additionally, the use of both SEM and Bootstrapping methods in our study serves to further validate the relationships between the variables, and the results from both methods converge, demonstrating that the path relationships in the model are stable and significant.

Comment 5:

I think that "Job Attribute Preferences" seems to be the most appropriate term to summarize items C1-C25, as it directly focuses on the job characteristics that students value. It's just a suggestion.

Response:

We sincerely appreciate your thoughtful suggestion regarding the terminology. After careful discussion and further consideration, we agree that the term "Job Attribute Preferences" is indeed highly appropriate as it accurately encapsulates the core content of items C1-C25, especially since these items directly reflect students' value of job characteristics. Consequently, we have revised the manuscript to ensure consistency and clarity in the use of this term, making the necessary adjustments throughout the text (please see lines 244-245 of the revised manuscript, as well as lines 781-782, Appendices B and C).

We would like to express our sincere gratitude once again to you and the reviewers for your invaluable feedback on our manuscript. Your suggestions have significantly enhanced the quality and rigor of our research. Should there be any further questions or areas requiring additional revisions, please do not hesitate to contact us.

Yours sincerely,

Huang Zerui

---

## [Editor Report · Decision Letter 3]

4 Feb 2025

PONE-D-24-47484R3The influence of psychological capital on employment expectations of vocational undergraduate students: The chain mediating role of active coping style and educational flow experiencePLOS ONE

Dear Dr. Huang,

Thank you for submitting your manuscript to PLOS ONE. After careful consideration, we feel that it has merit but does not fully meet PLOS ONE’s publication criteria as it currently stands. Therefore, we invite you to submit a revised version of the manuscript that addresses the points raised during the review process.

We look forward to receiving your revised manuscript.

Kind regards,

Dr. Manuel Salas-Velasco, PhD

Academic Editor

PLOS ONE

Journal Requirements:

**Additional Editor Comments:**

In the context of bootstrapping in SEM, the estimated values of the coefficients typically remain practically the same as the original estimates obtained from the model. However, what does change are the standard errors (SE) and confidence intervals (CI).

Therefore, I suggest that the authors remove Figures 2 and 3 and revise the manuscript's wording. Only as a theoretical model could Figure 1 remain. I believe that the information in Table 2 is sufficient to support the paths shown in Figure 1.

---

## [Author Response · Author response to Decision Letter 3]

5 Feb 2025

Response to editor

Dear Editor,

Thank you for your careful review and valuable feedback on our manuscript. We deeply appreciate your suggestions and have made comprehensive revisions to the paper accordingly. Below are our detailed responses to each of your comments:

Comment 1:

In the context of bootstrapping in SEM, the estimated values of the coefficients typically remain practically the same as the original estimates obtained from the model. However, what does change are the standard errors (SE) and confidence intervals (CI).

Therefore, I suggest that the authors remove Figures 2 and 3 and revise the manuscript's wording. Only as a theoretical model could Figure 1 remain. I believe that the information in Table 2 is sufficient to support the paths shown in Figure 1.

Response:

Thank you for your thoughtful suggestions regarding the use of bootstrapping in SEM and the feedback on the figures. We fully understand and agree with your point, particularly that while the estimated values of the coefficients typically remain unchanged in bootstrapping, the standard errors (SE) and confidence intervals (CI) do vary.

We concur with your suggestion that Figures 2 and 3 are not necessary in this context, as the information presented in Table 2 sufficiently supports the paths shown in Figure 1. In accordance with your recommendation, we have removed Figures 2 and 3 and made the necessary revisions to the manuscript, including the corresponding adjustments in the text (see revised manuscript, lines 312-326).

Additionally, the SEM-related terminology has been updated and modified throughout the manuscript, and the reference list has been adjusted for consistency (see references 69-88, lines 286, 300, 310, 314, 316, 351, 354, 356, 359, 368, 372, 377, 388, 390, 403, 406, 412).

Once again, we would like to express our sincere gratitude for your invaluable feedback, which has significantly improved the clarity and focus of our paper. We believe that these revisions have enhanced the overall quality of the manuscript, and we are grateful for your continued support throughout the process. Should you have any further questions or require additional modifications, please do not hesitate to contact us.

Sincerely,

Huang Zerui

---

## [Editor Report · Decision Letter 4]

7 Feb 2025

The influence of psychological capital on employment expectations of vocational undergraduate students: The chain mediating role of active coping style and educational flow experience

PONE-D-24-47484R4

Dear Dr. Huang,

We’re pleased to inform you that your manuscript has been judged scientifically suitable for publication and will be formally accepted for publication once it meets all outstanding technical requirements.

Kind regards,

Dr. Manuel Salas-Velasco, PhD

Academic Editor

PLOS ONE

**Additional Editor Comments:**

In my previous report, I did not mean that SEM should be removed because it is the technique used, only Figures 2 and 3 because they caused confusion with the results of Table 2. In lines 312 to 314 (new version), it should be rewritten as:

"To test the hypotheses, this study employed bootstrapping within a structural equation modeling (SEM) framework to examine the chain mediating model of active coping style and educational flow experience between psychological capital and employment expectations."

---

## [Editor Report · Acceptance letter]

PONE-D-24-47484R4

PLOS ONE

Dear Dr. Huang,

I'm pleased to inform you that your manuscript has been deemed suitable for publication in PLOS ONE. Congratulations! Your manuscript is now being handed over to our production team.

Kind regards,

on behalf of

Dr. Manuel Salas-Velasco

Academic Editor

PLOS ONE